# Association of Pro-Inflammatory Cytokines with Vitamin D in Hashimoto’s Thyroid Autoimmune Disease

**DOI:** 10.3390/medicina59050853

**Published:** 2023-04-28

**Authors:** Amer Siddiq, Abdul Khaliq Naveed, Nabila Ghaffar, Muhammad Aamir, Naveed Ahmed

**Affiliations:** 1Pathology Department, Islamic International Medical College, Riphah International University, Islamabad 46000, Pakistan; 2Department of Medical Education, Avicenna Medical College, Lahore 54000, Pakistan; 3Department of Medical Microbiology and Parasitology, School of Medical Sciences, Universiti Sains Malaysia, Kubang Kerian 16150, Malaysia

**Keywords:** interleukins, pro-inflammatory cytokines, Hashimoto’s thyroiditis, serum parameters

## Abstract

*Background and objectives:* Hashimoto’s thyroiditis is an important autoimmune thyroid condition. It is characterized by lymphocytic congestion of the thyroid gland followed by progressive deterioration and fibrous substitution of the thyroid in the parenchymal structure. This study has provided insight into the variations of blood pro-inflammatory cytokine levels in patients with Hashimoto’s disease and the key role of vitamin D levels among selected patients. *Materials and Methods:* A total of 144 participants including healthy controls and patients were studied in the current study in which 118 were female and 26 were male. The thyroid profile was evaluated in patients with Hashimoto’s thyroiditis and healthy controls. *Results:* The mean ± SD Free T4 in the patients was recorded as 14.0 ± 4.9 pg/mL, and TSH was 7.6 ± 2.5 IU/L, whereas the median ± IQR thyroglobulin antibodies (anti-TG) were 285 ± 142. Thyroid peroxidase antibodies (anti-TPO) were 160 ± 63.5, whereas in the healthy controls, the mean ± SD Free T4 was recorded as 17.2 ± 2.1 pg/mL, and TSH was 2.1 ± 1.4 IU/L, whereas the median ± IQR anti-TGs were 56.30 ± 46.06, and anti-TPO was 5.6 ± 5.12. The assessment of pro-inflammatory cytokines (pg/mL) and total Vitamin D levels (nmol/L) in patients with Hashimoto’s thyroiditis was recorded with values IL-1B 6.2 ± 0.8, IL-6 9.4 ± 0.4, IL-8 7.5 ± 0.5, IL-10 4.3 ± 0.1, IL-12 3.8 ± 0.5, TNF-α 7.6 ± 1.1, and total vitamin D 21.89 ± 3.5, whereas in healthy controls the mean ± SD IL-1B was 0.6 ± 0.1, IL-6 2.6 ± 0.5, IL-8 3.0 ± 1.2, IL-10 3.3 ± 1.3, IL-12 3.4 ± 0.4, TNF-α 1.4 ± 0.3 and total vitamin D was 42.26 ± 5.5. *Conclusions:* It was found that individuals with Hashimoto’s thyroiditis had raised serum levels of IL-1B, IL-6, IL-8, IL-10, IL-12, and TNF-α as compared to the healthy controls, whereas the total vitamin D levels were remarkably low as compared to health controls. Serum TSH, anti-TG, and anti-TPO levels were typically lower in controls and much higher in individuals with Hashimoto’s thyroiditis. The current study’s findings might aid in future studies and in the diagnosis and management of autoimmune thyroid disease.

## 1. Introduction

Thyroid diseases, regardless of gender or age, are among the most common glandular disorders worldwide [1]. These thyroidal disorders often range from subclinical with asymptomatic patients to symptomatic clinical ones [2]. Other common endocrine dysfunctions include subacute episodes, goiter, iodine deficiency disorders, Hashimoto’s thyroiditis (HT) [3], Graves’ disease (GD), and thyroid cancer [4]. These illnesses have been reported in 109 regions worldwide, putting 1.6 billion individuals at risk [5].

Hashimoto’s disease is an autoimmune disorder that can affect the thyroid gland and is also known as lymphatic thyroiditis [6,7], characterized by lymphocytic congestion of the thyroid gland followed by progressive deterioration and fibrous substitution of thyroid parenchymal structure [8]. Symptoms of Hashimoto’s disease can include fatigue, weight gain, cold intolerance, hair loss, joint pain, depression, and an enlarged thyroid gland (goiter). Over time, as the thyroid gland becomes more damaged, hypothyroidism can develop, leading to a reduction in the levels of thyroid hormones in the blood and goiter could or might not arise in patients [9].

The most clinically relevant anti-thyroid autoantibodies are thyroid peroxidase antibodies (anti-TPO) and thyroglobulin [10]. There is little evidence that both antibodies have a significant involvement in the pathophysiology of Hashimoto’s thyroiditis, and it is much more likely that the disease will develop as a result of T-cell-mediated cytotoxicity and the activation of apoptotic pathways [11]. Hashimoto’s disease is usually diagnosed with blood levels of thyroid hormones and thyroid-stimulating hormones (TSH). An elevated TSH level and a low level of thyroid hormones can indicate hypothyroidism, which is a common finding in Hashimoto’s disease. Nevertheless, Tabs are also a helpful sign for Hashimoto’s thyroiditis autoimmunity diagnosis [12]. Nearly all patients (>90%) have anti-TPO, whereas only about (80%) of patients have anti-TGs [13]. The prevalence of subacute hypothyroidism in Pakistan is 4.1 and 5.4%, respectively [14].

Vitamin D prevents the production of some inflammatory cytokines, including tumor necrosis factor-α (TNF-α), interleukin (IL)-1, IL-6, IL-8, and IL-12. Sufficient amounts of vitamin D suppress the differentiation and development of dendritic cells because the production of MHC class II proteins, co-stimulatory molecules, and IL-12 is reduced [15]. Another factor is vitamin D’s ability to influence the activity of regulating T cells, which reduces the impact of T-cell-dependent immunity in autoimmune disorders. Specifically, T cells and B cells react to thyroid antigens in individuals with a hereditary predisposition, which can result in the development of hyperthyroidism. Increased production of thyroid antigens such as anti-TPO and anti-thyroglobulin was associated with serum concentrations of vitamin D3 [16].

Cytokines are essential for regulating immune reactions that control the equilibrium between the start of autoimmunity and the preservation of self-tolerance. However, cytokines’ function is frequently unclear and neither autonomous nor exclusive of other immunological molecules [17]. A regulating hormone may favor the initiating and/or worsening autoimmune reactions or it may favor the developing resistance against autoimmune thyroid disease [18]. The type of co-signaling provided by other cytokines mainly affects these seemingly opposing activities of a particular cytokine. Therefore, the creating of suitable methods to regulate cytokine responses to preserve or recover health depends on a comprehensive knowledge of a particular cytokine’s function in the setting of a particular immunological reaction [19].

Since there is no effective management for Hashimoto’s thyroiditis, thyroxine replacement therapy is mostly used to manage hypothyroidism (oral levothyroxine level four) [20]. Despite the fact that the precise cause of Hashimoto’s thyroiditis is still not entirely understood, cellular and humoral immunity are crucial in the progression of this condition [21]. There is also evidence of an association between vitamin D levels and Hashimoto’s disease. Low levels of vitamin D have been observed in some people with Hashimoto’s disease, and some studies have suggested that blood variations of vitamin D level can change the levels of pro-inflammatory cytokines in the patients who have symptoms of Hashimoto’s disease. The current study was conducted to find out the association of pro-inflammatory cytokines with vitamin D levels in patients with Hashimoto’s disease.

## 2. Materials and Methods

### 2.1. Study Area

The current study was conducted by the Pathology Department, Islamic International Medical College, Riphah International University, Islamabad. An ethical approval was obtained before starting the study, collecting samples from the patients, and accessing the medical record of patients.

### 2.2. Subjects’ Selection Criteria

A total of 144 subjects were selected for the current study in which 72 were healthy controls and 72 were Hashimoto thyroiditis patients who were visiting various hospitals and endocrine facilities within the boundaries of Rawalpindi/Islamabad Pakistan. The selection criteria for the recruitment of healthy controls in the current study were to check their medical history and then the current health status. The healthy controls were mostly the hospital staff, university colleagues, and friends. If any of the healthy controls had previously suffered from any hepatic problems (e.g., hepatitis), calculus in the renal system, diabetes or hypertension, they were not recruited for the current study. At first, the control group participants and the patients registered in outdoor department were elaborated with the aim and objective of the study and then their permission for enrolling in the study were obtained. Written informed consent forms were obtained from all of the study participants accordingly.

#### 2.2.1. Ethical Considerations

The protocols set by the institutional review committee of Islamic international Medical College were followed while conducting the research and the rights of the research participants were:Permission of the competent authority was obtained before the start of the study;Written informed consent (attached) was taken from all the participants in accordance with provisions of the Declaration of Helsinki;All information and data collection were kept confidential;Participants remained anonymous throughout the study and their privacy was protected. Their identity will not be revealed in any publication resulting from this study;Participation in the case-control study was considered voluntary for the study subjects and their discretion not to participate or withdraw their consent at any time was observed. They were not penalized in any way if they decided not to participate or to withdraw from the study;Data about the study will not be shared with anyone except for academic purposes.

#### 2.2.2. Statement of Confidentiality

Confidentiality makes sure that subsequent investigations or scientific studies do not contain such personally identifiable data. Provided the consensus initiatives frequently include tiny numbers, it ought to be crucial to think carefully about how assessments are written to make absolutely sure that, even though identities were not used, there is no way for individuals to be recognized.

### 2.3. History and Medical Records

The study used patient history and medical records, previous medical scripts and documentations, marital status, gender, and age to analyze the autoimmune disease during the study with the help of concerned medical staff and patients.

### 2.4. Specimen Collection, Transportation, and Analysis

The whole blood samples were collected in EDTA containing vials. Blood samples of all the included patients with Hashimoto’s thyroiditis, as well as from the healthy controls, were collected in 0.5 M EDTA tubes for interleukins. For a total vitamin D analysis and thyroid profile including anti-thyroid autoantibodies, we used yellow capped tubes for the collection and storage of serum.

Interleukin-1B, Interleukin-6, Interleukin-8, Interleukin-10, Tumor Necrosis Factor, and Interleukin-12p protein concentrations were accurately measured in one specimen using the BD-CBA (Inflammatory Cytokines Kit). As per the specimen type, the performance of the kit was tuned for the measurement of particular protein types in supernatants of tissue culture, serum, and EDTA plasma specimens [22]. Testing for anti-thyroids and auto-antibodies was performed by ELISA, whereas testing for thyroid profile and total vitamin D levels was performed by the chemiluminescence technique. The total vitamin D analysis was performed by using an ADVIA Centaur(R) assay in vitro for diagnostic purposes on the ADVIA Centaur System.

In order to identify analytes using flow cytometry, BD CBA assays offer a technique for capturing a soluble analyte or group of analytes with beads of defined size and fluorescence. A particular antibody was coupled to each of the kit’s capture beads. Phycoerythrin (PE)-conjugated antibodies are the detection reagent included in the kit; they produce a fluorescent signal in proportion to the amount of bound analyte. Sandwich complexes (capture bead + analyte + detection reagent) are created when the capture beads and detector reagent are incubated with an unknown sample that contains known analytes. In order to identify particles with fluorescence properties of both the bead and the detector, these complexes can be evaluated using flow cytometry.

### 2.5. Statistical Consideration

Data entry and analysis were performed using SPSS version 21.0. The mean and S.D for quantitative variables such as free T4 (FT4), Thyrotropin (TSH), IL-10, IL8, IL-6, IL-1B, and IL-12 were calculated. Frequencies were calculated for qualitative variables. The *p*-value at ≤0.05 was considered statistically significant.

## 3. Results

A total of 144 subjects were recruited for the current study. The proportion of females was higher (*n* = 118) as compared to male (*n* = 26). A total of 72 (50%) subjects were the control group (healthy people) and 72 (50%) were the patients. A detailed description of the study subjects according to age group is presented in Table 1.

### 3.1. Levels of Thyroid Hormones and Thyroid Auto Antibodies in Patients with Hashimoto’s Diseases v/s Healthy Controls

The levels of TSH, anti-TPO, and Anti-TG were significantly higher in patients with Hashimoto’s disease than the healthy controls (*p* < 0.05), whereas there was no significant difference was seen for the levels of FT4 and FT3 among healthy controls and patients. A detailed description of thyroid profile and the level of thyroid auto antibodies is presented in Table 2.

### 3.2. The Concentrations of Pro-Inflamatory Cytokines in Patients with Hashimoto’s Diseases v/s Healthy Controls

The concentrations of IL-8, 10, IL-1B, and TNF-α were significant higher in patients with Hashimoto’s disease (*p* < 0.05), whereas the concentrations of IL-6 and 12 were non-significant among patients with Hashimoto’s disease as compared to healthy controls (*p* > 0.05). A detailed description of pro-inflamatory and anti-inflamatory cytokines among the studied groups has been given in Table 3.

### 3.3. The Total Vitamin D Levels in Patients with Hashimoto’s Disease v/s Healthy Controls

The results of the current study showed important findings: the total vitamin D levels in patients with Hashimoto’s disease was significantly lower, i.e., less than 25 nmol/L as compared to healthy control group values which were 42.26 nmol/L; however, the results were still lower than the normal reference range with a significant *p*-value of 0.001 (Figure 1).

### 3.4. Correlation between Anti-TPO and Total Vitamin D Levels

The correlation between anti-TPO and total vitamin D levels among healthy controls and patients has been shown in Figure 2. Anti-TPO correlation with total vitamin D level was significant, which showed that the anti-TPO level was significantly increased in Hashimoto patients compared to the control. The control group showed a uniform distribution of anti-TPO and vitamin D level correlation, whereas Hashimoto’s anti-TPO increased up to 500 mmol/IU.

### 3.5. Correlation between Anti-TG and Total Vitamin D Levels

The correlation between anti-TG levels and total vitamin D levels among healthy controls and patients is presented in Figure 3.

Anti-TG levels and vitamin D levels in the control group were not positively significant, whereas in patients with Hashimoto’s disease, both showed a significant correlation and levels raised up to 500.

### 3.6. Correlation between Pro-Inflamatory Cytokines and Total Vitamin D Levels

The correlation between TNF-α and total vitamin D levels among healthy controls and patients is presented in Figure 4.

The correlation between TNF-α and total vitamin D level in the control group was uniform and the TNF-α level was not more than 2.0, whereas in case of Hashimoto’s patients, the TNF-α level increased up to 10.8. There was a significant difference observed between the concentrations of Interleukins 8, 10, and 1B when correlated with the total vitamin D level of healthy controls and patients (*p* < 0.05), whereas no significant difference was observed between the IL-6 and 12 when compared to vitamin D levels (*p* > 0.05) in patients with Hashimoto’s disease.

## 4. Discussion

One of the most prevalent organ-specific autoimmune disorders is autoimmune thyroid disease. The HT and GD are two of the most prevalent clinical manifestations of autoimmune thyroid disease. Hypothyroidism caused by an inflammatory reaction mediated by T cells specific to thyroglobulin characterizes HT [9]. In comparison, GD is marked by hyperthyroidism caused by increased thyroid hormone synthesis brought on by stimulatory antigens that are particular for the thyrotropin receptor [6]. Vitamin D deficiency is becoming higher all over the world. Over the last three decades, it has become evident that vitamin D has a role in more than simply regulating calcium homeostasis and preserving bone health. One of the important extra-skeletal effects of vitamin D controlling the immune system. Several autoimmune disorders, including HT, have been associated with vitamin D deficiency. The relationship between vitamin D deficiency and HT is documented but still requires further studies. Hence, the current study aimed to see the vitamin D levels in relation to pro-inflammatory cytokines in HT patients and healthy controls.

The current study reveals the negative relationship of pro-inflammatory cytokine levels with total vitamin D levels in HT patients, i.e., with the decreased total vitamin D levels, the inflammatory cytokine levels start increasing. The proinflammatory cytokine concentrations of IL-8, IL-10, IL-1B, and TNF-α were found to be significantly increased in patients with Hashimoto’s disease, who have anti-thyroid peroxidase antibodies and/or anti-thyroglobulin antibodies as compared to age- and sex-matched controls. Data from a recent study showed that irrespective of the functional status of the disease, blood IL-12 levels were elevated in all subgroups of HT patients. Similarly, the second interleukin was studied, and it was shown that hypothyroid HT patients had a substantial increase in the serum levels of IL-10, compared to euthyroid HT and controls [18]. However, in the present study, the IL-12 was not significantly changed in patients with Hashimoto’s disease as compared to IL-10, which was also increased. Moreover, screening the blood levels of IFN-α and IL-8 can help distinguish between patients with thyroid illnesses and healthy persons. IL-8 appears to have a role in the etiology of thyroid diseases and may represent a target for improved diagnostic and therapeutic techniques [23].

Total vitamin D is critical in immunomodulation and tends to reduce intracellular reactive oxygen species. A low total vitamin D level could be a reason for induction of inflammation in patients with rheumatoid arthritis. The role of vitamin D in oxidation-reduction signaling indicates that low 25 (OH) total vitamin D enhances the production of ROS. Such chemically active species can react with NO_2_ to form peroxynitrite which is a highly reactive compound that may lead to depletion of oxidation biomolecules and antioxidants. Redox balance could be disturbed as a result signaling cascade (NF-κB) becoming activated, which as a result stimulates the cytokines formation in patients with rheumatoid arthritis [16]. It has been established by Mateen S et al. that 25 (OH) vitamin D levels are in inverse correlation with ROS and positively correlate proinflammatory cytokines with RA disease severity in patients with seropositive rheumatoid arthritis [24]. However, in our study, it is shown that serum levels of cytokines are increased in HT patients as compared to controls, and the level of total vitamin D is decreased in our patient group as compared to normal healthy individuals.

Nozawa K et al., suggested that TNF-α is the main cytokine that modulates the genesis of other inflammatory cytokines [25]. This finding suggests that TNF-α levels are supposed to be increased during active inflammation, which is the case in our study where HT patients have increased TNF-α levels compared to the controls. IL-1β and TNF-α are the known necrosis factor κB cascade stimulators and end with positive feedback stimulus.

IL-6 plays an essential role in thyroid disease progression, and by IL-6 signaling, thyroid disease can be managed [26]. The transcription of inflammatory mediators is enabled by the translocation of NF-B dimers to the nucleus due to TNF-’s assistance in the phosphorylation of inhibitor kinase beta. IL-1β and TNF-α exhibit similar pro-inflammatory effects, such as the activation of T cells and the promotion of the synthesis of inflammatory cytokines [27]. Our findings are the same as the fact that IL-1B was increased compared to healthy controls. Another earlier study related to this found that patients with benign thyroid disorders and patients with PTC had considerably greater blood TNF-α [28].

Yoshida Y reported that elevated IL-6 levels are present in the synovial and blood of rheumatoid arthritis patients. Its actions on neutrophils cause the release of reactive oxygen intermediates and proteolytic enzymes, thus aiding joint destruction and inflammation. These cytokines are associated with pathological investigations such as increased C-reactive protein and coagulation ability and decreased albumin levels. These also have significant associations with inflammatory markers in rheumatoid arthritis patients. More likely, it is known as a stronger immunoglobulins stimulator and more responsible for acute phase protein production than TNF-α or IL-1B. Serum levels of IL-6 have also positively correlated with TNF-α [29]. In our study, IL6 is high in HT patients compared to normal controls; however, the results are non-significant. IL-10 is an immunoregulatory and anti-inflammatory cytokine that can decrease the antigen-presenting cells performance and the production of all inflammatory cytokines. IL-10 activates the synthesis of monocytes’ tissue inhibitor of metalloproteinases-1 (TIMP-1) and also stops the synthesis of proteases. An earlier study found that patients with benign thyroid disorders and patients with PTC had considerably greater blood TNF-α [30]. According to previous research findings, TSH levels were significantly higher in the group that had HT. The fact that TSH is known to increase the production of inflammatory substances also lends credence to the hypothesis that TSH may be linked to the development of HT [31]. Moreover, TSH is considered a potent stimulator of IL-10 and IL-12 which may end with an increased level of both interleukins [32]. However, indications in this research are quite different because increased TSH in patients does not result in increased IL-12.

Van Roon JA et al. have shown that the raised concentration of detection of interleukin 10 in the plasma of RA patient agrees with previous studies in the later stage of the disease [33]. In our study, IL10 was also significantly increased as compared to normal individuals. IL-12 is a potent, pro-inflammatory cytokine synthesized by antigen-presenting cells typically in response to microbial pathogens. IL-12 is mainly responsible for the initiation and boosting of cell-mediated immunity. Nguyen KG demonstrated the role of IL-12 as an excellent antitumor cytokine against a range of malignancies in preclinical studies [15]. In our study, the level of IL-12 was found to be slightly raised in patients with Hashimoto’s disease than in healthy patients, but this was statistically insignificant.

In many of cell types, the formation of IL-8 is strongly excited by IL-1 and TNF-α. Leukoregulin increases the expression of IL-8 fibroblasts in the human skin because, as per previously published data about IL-6, IL-8 and TNF-α cytokines have a significant role in the management and prognosis of tumors [19]. Additionally, phytohemagglutinins, double-stranded RNA, concanavalin A, sodium urate crystals, phorbol esters, bacterial lipopolysaccharides, and viruses induce the production of IL-8. IL-7 increases the expression of IL-8 in both resting and activated human blood monocytes. In chondrocytes, the formation of IL-8 is excited by IL1-β, and TNF-α. In human astrocytes, the formation and secretion of IL-8 is stimulated by IL-1 and TNF-α. Qazi BS has shown that the synthesis of IL-8 is inhibited by glucocorticoids, IL-4, TGF-, inhibitors of 5′ lipoxygenase, and 1.25 (OH) 2 vitamin D3. Different carcinoma cell lines constitutively and often synthesize IL-8, and its synthesis may be connected to the rise of serum IL-8 in individuals with liver cancer. In epithelial, fibroblastic cells, endothelial cells secretion of IL-8 is stimulated by IL-17 [34]. In our study, IL-8 was significantly increased in HT patients compared to the controls.

Study limitations: Although the current study reported significant results in terms of antibodies, hormones, cytokines, and total vitamin D levels, it also had some limitations which can add more impact in the study. Because of the financial and other limitations in the research facilities, the T-cell subtypes, and the anti-inflammatory cytokines could not be studied for more comparison between the studied groups.

## 5. Conclusions

We determined that patients with Hashimoto’s thyroiditis had substantial levels of IL-1B, IL-8, and IL-12 and TNF-α, but that the levels of IL-6 and IL-12 were non-significant compared to the healthy control group. Serum TSH, anti-TG, and anti-TPO levels were typically lower in controls and much higher in patients with Hashimoto’s thyroiditis. Moreover, there were significantly low levels of total vitamin D levels seen in patients with Hashimoto’s thyroiditis who showed a severe deficiency. The current study’s findings will aid in future scientific testing, as well as the diagnosis and management of autoimmune thyroid diseases.

## Figures and Tables

**Figure 1 medicina-59-00853-f001:**
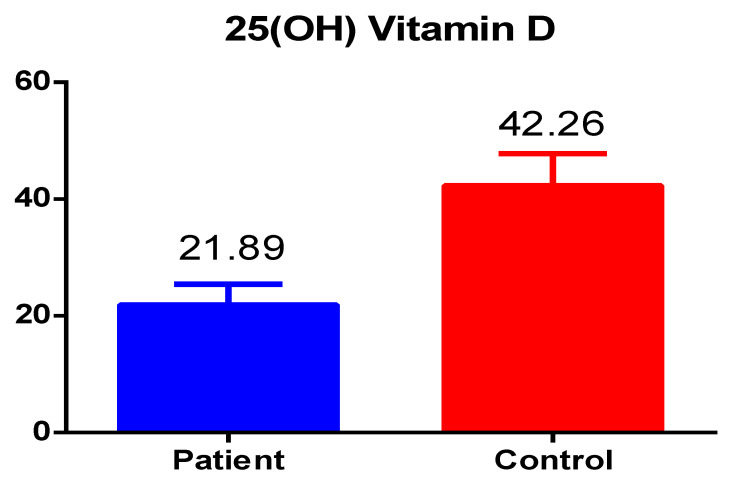
The total vitamin D levels in patients and healthy controls.

**Figure 2 medicina-59-00853-f002:**
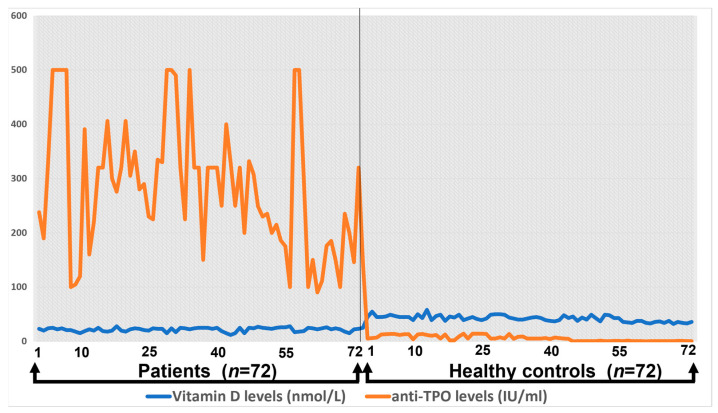
The correlation of total vitamin D levels with anti-TPO.

**Figure 3 medicina-59-00853-f003:**
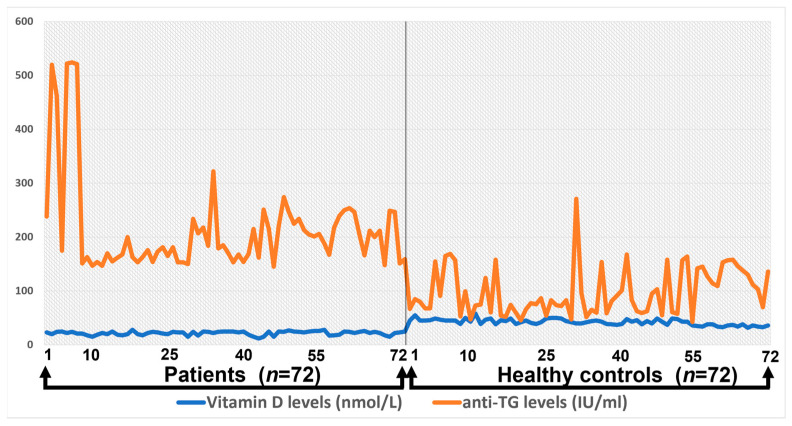
The correlation between anti-TG levels and total vitamin D levels among healthy controls and patients.

**Figure 4 medicina-59-00853-f004:**
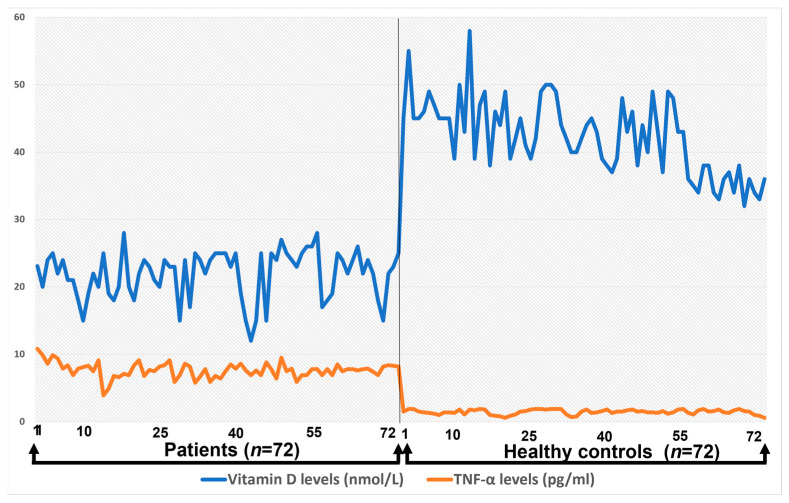
The correlation between TNF-α and total vitamin D levels among healthy controls and patients.

**Table 1 medicina-59-00853-t001:** Age group-wise distribution of study participants.

Age	*n*	%
18–25	27	18.7
26–30	28	19.4
31–35	32	22.2
36–40	23	15.9
41–45	11	7.6
46–50	8	5.5
51–55	13	9
56–60	2	1.3

Vitamin D serum level showed significant correlation with TNF-α, as vitamin D levels decrease in Hashimoto’s disease, the level of TNF-α increases.

**Table 2 medicina-59-00853-t002:** Levels of thyroid hormones and thyroid auto antibodies.

Variables	Unit	Hashimoto’s Disease	Control
TSH	mIU/L	7.674 ± 2.55 (M ± SD)	2.13 ± 1.42 (M ± SD)
FT4	pmol/L	14.03 ± 4.96 (M ± SD)	17.25 ± 2.16 (M ± SD)
FT3	pg/mL	2.69 ± 0.19 (M ± SD)	2.80 ± 0.15 (M ± SD)
Anti-TPO	(IU/mL)	285 ± 142 Median ± IQR	5.0 ± 10.69 Median ± IQR
Anti-TG	(IU/mL)	160 ± 63.5 Median ± IQR	39.6 ± 80.05 Median ± IQR

**Table 3 medicina-59-00853-t003:** Levels of pro-inflammatory and anti-inflammatory cytokines in the healthy controls and patients with Hashimoto’s disease.

Variables	Unit	Hashimoto’s Disease	Healthy Controls	*p*-Value
Interlukin-1B	pg/mL	6.253 ± 0.382	0.601 ± 0.1756	0.000 *
Interlukin-6		9.465 ± 0.486	2.632 ± 0.553	0.393
Interlukin-8		7.539 ± 0.558	3.063 ± 1.235	0.008 *
Interlukin-10		4.325 ± 0.497	3.396 ± 0.353	0.002 *
Interlukin-12		3.813 ± 0.483	3.44 ± 0.481	0.850
TNF-α		7.662 ± 1.132	1.459 ± 0.357	0.000 *

* statistically significant.

## Data Availability

The data related to the current study can be accessed upon a reasonable request to the corresponding author.

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
