# Peer review of "Association of Pro-Inflammatory Cytokines with Vitamin D in Hashimoto’s Thyroid Autoimmune Disease"

_medicina, 2023, doi:10.3390/medicina59050853_

Round 1

Reviewer 1 Report

Congratulations to the authors for their work.

The data obtained regarding the number of samples collected and the measured cytokines will contribute to the literature.

The summary, Introduction, and methodological details are clear enough.

The presentation of the results section should be improved.

Comments:

The large number of patient and control samples evaluated within the study's scope makes the results efficiently. However, the patient group's epidemiological and clinical features should be shared. More than table 1 is needed in this regard. In terms of detail, including the age of diagnosis, the duration of the treatment they receive, etc., that will increase the effectiveness of the study. For 72 patient samples, I suggest adding them beside the classification chart in table 1. 

Please select different patient and control group icons for the correlation graphs and indicate them in the legends section.

For Figure 4, add the p-value for statistical significance.

In the discussion part, I understand the comparison of literature support with RA regarding cytokines. However, more significance can be obtained in studies on HT, in review articles, in terms of cytokines and the T cell subtypes they act on.

It would be more effective if the study included both pro-inflammatory and anti-inflammatory cytokines and the differences in their levels compared to the control with T cells and HT. Such references below

PMID: 36509987 and/or PMID: 36336870 and/or PMID: 25971541.

*There could be more meaning in the outcomes. Examining them in the discussion section will increase the effect of the study. Cytokines have synergistic effects, and these effects have many aspects in autoimmune diseases. It is possible to discuss the epidemiology and differences from the healthy group more efficiently by examining the T cell subtypes, especially regarding pro- and anti-inflammatory cytokines. 

Author Response

Reviewer 1

Comments and Suggestions for Authors

Congratulations to the authors for their work. The data obtained regarding the number of samples collected and the measured cytokines will contribute to the literature. The summary, Introduction, and methodological details are clear enough. The presentation of the results section should be improved.

Response: Dear reviewer, we would like to thank you for your kind appreciation to the present study. Thank you for your valuable comments and suggestions. We have revised the results section in the revised version of manuscript.

Comments:

The large number of patient and control samples evaluated within the study's scope makes the results efficiently. However, the patient group's epidemiological and clinical features should be shared. More than table 1 is needed in this regard. In terms of detail, including the age of diagnosis, the duration of the treatment they receive, etc., that will increase the effectiveness of the study. For 72 patient samples, I suggest adding them beside the classification chart in table 1. 

Response: Dear reviewer, thank you for your valuable suggestion to include the epidemiological and clinical features in order to make the manuscript more suitable for the reader. All of the patients studied in the present study were newly diagnosed and were not started with the treatment yet. Hence, the data regarding the duration of diagnosis and treatment can not be added. Furthermore, as the patients were belonging to one center, we can not add the epidemiological data in this regard. That’s why we would like to request you to allow us not to add this data and proceed as it is.

Please select different patient and control group icons for the correlation graphs and indicate them in the legends section.

Response: Dear reviewer, thank you for your valuable suggestion. The former graphs were generated from SPSS where no editing option was available and we preferred to not to use the image editing software. Considering your suggestion, we have revised the graphs using Microsoft Excel, which are simple but more understandable.

For Figure 4, add the p-value for statistical significance.

Response: Line 174: p-value has been added.

In the discussion part, I understand the comparison of literature support with RA regarding cytokines. However, more significance can be obtained in studies on HT, in review articles, in terms of cytokines and the T cell subtypes they act on. It would be more effective if the study included both pro-inflammatory and anti-inflammatory cytokines and the differences in their levels compared to the control with T cells and HT. Such references below

PMID: 36509987 and/or PMID: 36336870 and/or PMID: 25971541.

Response: Line 326-330: Dear reviewer, thank you for your kind suggestion to improve the discussion section and to add more parameters in the study. We acknowledge that the suggestion will add a very significant impact to the study, but after seeing the current facilities, we are not able to run this further analysis. Hence, a study limitations section has been added at the end of discussion section. We have revised the discussion section and discussed the results with more previous studies. Following new references have been added:

  1. Wu J., Huang H., Yu X., How does Hashimoto's thyroiditis affect bone metabolism?, Reviews in endocrine & metabolic disorders 2023, 24, 191-205, doi:10.1007/s11154-022-09778-x.
  2. Jin B., Wang S., Fan Z., Pathogenesis Markers of Hashimoto's Disease-A Mini Review, Frontiers in bioscience (Landmark edition) 2022, 27, 297, doi:10.31083/j.fbl2710297.
  3. Zivancevic-Simonovic S., Mihaljevic O., Majstorovic I., Popovic S., Markovic S., Milosevic-Djordjevic O., Jovanovic Z., Mijatovic-Teodorovic L., Mihajlovic D., Colic M., Cytokine production in patients with papillary thyroid cancer and associated autoimmune Hashimoto thyroiditis, Cancer immunology, immunotherapy: CII 2015, 64, 1011-1019, doi:10.1007/s00262-015-1705-5.

*There could be more meaning in the outcomes. Examining them in the discussion section will increase the effect of the study. Cytokines have synergistic effects, and these effects have many aspects in autoimmune diseases. It is possible to discuss the epidemiology and differences from the healthy group more efficiently by examining the T cell subtypes, especially regarding pro- and anti-inflammatory cytokines. 

Response: Dear reviewer, we would like to appreciate your kind suggestion to add more aspects in the current study specially the T cell sub types and anti-inflammatory cytokines. We acknowledge that the suggestion will add a very significant impact to the study, but after seeing the current facilities, we are not able to run this further analysis. As our study did not received any funding from any of the institution, and all of the expenses has been taken at our own. I hope that you will understand our situation, as because of the financial limitations we can not add more parameters in the current study. But we will keep these suggestion in mind while conduction another study in near future. We will really appreciate if you allow us to not to include any further parameters in the current study. We have added a study limitation section at the end of discussion section. Furthermore, we have revised the discussion section and added more reference studies.

Reviewer 2 Report

Paper “Association of Pro-inflammatory Cytokines with Vitamin D in Hashimoto's Thyroid Autoimmune Disease” tries to highlight importance of vitamin D level in autoimmune thyroiditis development.

Simple and adequate methodology was used to achieve this study’s objective, which was to find out the association of pro-inflammatory cytokines with Vitamin-D level in Hashimoto’s disease patients. However, the results were poorly interpreted and structured illogically. For example, correlation results between vitamin D level and TNF-α, anti-TPO, and anti-TG were put into the beginning of “Results” section under “3.1. Demographic characteristics” subtitle, which is very awkward and does not reflect its content. In addition, correlation with other cytokines are mentioned only in one sentence afterwards. Since the aim of this study was to find out the association of pro-inflammatory cytokines with Vitamin-D level all correlations (including all enrolled in this study) cytokines should be placed at the end of “Results” section. The comparison of hormones’, auto-antibodies’ and cytokines’ levels between studied groups should be placed at the beginning. Comparison of hormones’, auto-antibodies’ and cytokines’ levels are better to present as dot blot graphs with appropriate statistical method.

Levels of auto-antibodies presented in this paper has too big standard deviation’s values, which indicates that not all data are normally distributed. Therefore, all data should be checked for normal distribution. Non-normally distributed data should be presented as median±IQR instead of mean±SD.

English used in this paper is very weak and should be completely revised. Some sentences are very strange, for example, Lines 17-18, in abstract section. Some missing words and articles are present in the text. In addition, some of abbreviations should be revised – change TNF to TNF-α, TPO and TG antibodies could be shortened as anti-TPO and anti-TG. 

Author Response

Reviewer 2

Comments and Suggestions for Authors

Paper “Association of Pro-inflammatory Cytokines with Vitamin D in Hashimoto's Thyroid Autoimmune Disease” tries to highlight importance of vitamin D level in autoimmune thyroiditis development.

Simple and adequate methodology was used to achieve this study’s objective, which was to find out the association of pro-inflammatory cytokines with Vitamin-D level in Hashimoto’s disease patients. However, the results were poorly interpreted and structured illogically. For example, correlation results between vitamin D level and TNF-α, anti-TPO, and anti-TG were put into the beginning of “Results” section under “3.1. Demographic characteristics” subtitle, which is very awkward and does not reflect its content. In addition, correlation with other cytokines are mentioned only in one sentence afterwards. Since the aim of this study was to find out the association of pro-inflammatory cytokines with Vitamin-D level all correlations (including all enrolled in this study) cytokines should be placed at the end of “Results” section. The comparison of hormones’, auto-antibodies’ and cytokines’ levels between studied groups should be placed at the beginning. Comparison of hormones’, auto-antibodies’ and cytokines’ levels are better to present as dot blot graphs with appropriate statistical method.

Response: Dear reviewer, we would like to thank you for your kind appreciation to the present study. Thank you for your valuable comments and suggestions. We have revised the results section in the revised version of manuscript. Furthermore, we have rearranged the results section according to your suggestion. The subheading “Demographic characteristics” has been removed. We have tried to make the graphs for the comparison of hormones, auto-antibodies and the cytokines rather than the tables, but we found that the representation in form of tables was more understandable as compare to the dot blot graphs. Hence, we would like to request you to allow us to keep the information in form of tables and not to add the graphs.

Levels of auto-antibodies presented in this paper has too big standard deviation’s values, which indicates that not all data are normally distributed. Therefore, all data should be checked for normal distribution. Non-normally distributed data should be presented as median±IQR instead of mean±SD.

Response: Line 23, table 2: Dear reviewer, the values for anti-TG and anti-TPO has been changed as Median±IQR.

English used in this paper is very weak and should be completely revised. Some sentences are very strange, for example, Lines 17-18, in abstract section. Some missing words and articles are present in the text. In addition, some of abbreviations should be revised – change TNF to TNF-α, TPO and TG antibodies could be shortened as anti-TPO and anti-TG. 

Response: Dear reviewer, thank you for your valuable suggestion to revise the manuscript for English proofreading. The revised version of manuscript has been thoroughly revised for English proofreading and grammatical mistakes by a native English speaker. Furthermore, the abstract has been thoroughly revised and the sentence structures has been corrected. The manuscript has been checked for abbreviations, and the abbreviations has been placed at their first appearance. The TNF has been replaced with TNF-α, TPO and TG with anti-TPO and anti-TG throughout the manuscript.

Round 2

Reviewer 1 Report

Graphs presenting it in Excel instead of correlation graphs in the result section have increased the understandability of the data. The details of the discussion section have been improved with current references. In particular, the results about the level of cytokines were examined in more detail. Many corrections about English editing are highly recommended. 

Author Response

Reviewer 1

Comments and Suggestions for Authors

Graphs presenting it in Excel instead of correlation graphs in the result section have increased the understandability of the data. The details of the discussion section have been improved with current references. In particular, the results about the level of cytokines were examined in more detail. Many corrections about English editing are highly recommended. 

Response: Dear reviewer, thank you for your valuable suggestion and recommendations. We must appreciate that after addressing the comments from you and other reviewers, the quality of our manuscript has been significantly improved. The recent corrections suggested by you on the pdf file have been addressed in the revised version of the manuscript.

Reviewer 2 Report

Although, the manuscript was sufficiently improved there are still some critical questions:

·       The number of enrolled in study individuals is not consistent. In methods is indicated total number of 142 individuals, on other hand authors are indicating that group is consisted from 72 patients and 72 controls (which give totally of 144 individuals). In first table (Age group-wise distribution) 143 individuals are reflected. Also, in this table column “f” what does mean? For number of individuals usually “n” used as abbreviation;

·       Line 173 and Table 2 – “Thyroid profile” better to replace on Thyroid hormones levels (or something similar);

·       Table 2, row of “FT3”- “Pg” are indicated with upper case. Also, in this row mean level (and SD) of FT3 hormone is the same for both patients and controls. May be this is a mistake? How mean values and standard deviation can be equal for two groups?

·       Title of table 3 (“Levels of pro-inflammatory cytokines and TNF-α …”) is impropriate and should be changed, as a TNF-α is also a pro-inflammatory cytokine, in its turn, IL-10 is an anti-inflammatory cytokine;

New figures are less representative, especially Figure 1. In this Figure there no visible difference between anti-TPO levels in patients vs controls. Previous figures were better, however, authors have problems to edit legends to distinguish patients’ data from controls’. Therefore, as suggestion, authors could make figure for each group and make a compilation. Or to choose another type of figures, for example, column with dots.

Author Response

Reviewer 2

Comments and Suggestions for Authors

Although, the manuscript was sufficiently improved there are still some critical questions:

  • The number of enrolled in study individuals is not consistent. In methods is indicated total number of 142 individuals, on other hand authors are indicating that group is consisted from 72 patients and 72 controls (which give totally of 144 individuals). In first table (Age group-wise distribution) 143 individuals are reflected. Also, in this table column “f” what does mean? For number of individuals usually “n” used as abbreviation;

Response: Dear reviewer, thank you so much for your valuable validation and highlighting the typing error in the manuscript. The total number of participants included in the current study were 144. The numbers have been corrected wherever needed. Line 17-18, 99, 165-166. In table 1, the “f” has been replaced with “n”.

  • Line 173 and Table 2 – “Thyroid profile” better to replace on Thyroid hormones levels (or something similar);

Response: Line 172, 179: Thank you for your valuable suggestion. The change has been made and highlighted in red colour.

  • Table 2, row of “FT3”- “Pg” are indicated with upper case. Also, in this row mean level (and SD) of FT3 hormone is the same for both patients and controls. May be this is a mistake? How mean values and standard deviation can be equal for two groups?

Response: Table 2: The “Pg” has been changed as “pg”. The mean and SD values for FT3 has been corrected.

  • Title of table 3 (“Levels of pro-inflammatory cytokines and TNF-α …”) is impropriate and should be changed, as a TNF-α is also a pro-inflammatory cytokine, in its turn, IL-10 is an anti-inflammatory cytokine;

Response: Line 187, Table 3: The table legend has been corrected.

New figures are less representative, especially Figure 1. In this Figure there no visible difference between anti-TPO levels in patients vs controls. Previous figures were better, however, authors have problems to edit legends to distinguish patients’ data from controls’. Therefore, as suggestion, authors could make figure for each group and make a compilation. Or to choose another type of figures, for example, column with dots.

Response: Dear reviewer, thank you for your valuable suggestion to improve the figures. We have revised the figure 2, 3 and 4. We think that the figures became more better for the reader now. Furthermore, we have tried to explain the healthy controls and patients in the figures in order to make it clearer for the reader.